# The Cap and the Spermatostyle Protecting the Sperm Bundle Have a Similar Origin—Ultrastructural Study of the Spermatogenesis from the Ground Beetle *Carabus (Chaetocarabus) lefebvrei* Dejean, 1826 (Adephaga Carabidae)

**DOI:** 10.3390/insects15110864

**Published:** 2024-11-05

**Authors:** Pietro Lupetti, David Mercati, Anita Giglio, Pietro Brandmayr, Romano Dallai

**Affiliations:** 1Department of Science Life, University of Siena, 53100 Siena, Tuscany, Italy; pietro.lupetti@unisi.it (P.L.); david.mercati@unisi.it (D.M.); 2Department of Biology, Ecology and Earth Sciences, Di.B.E.S.T., University of Calabria, 87036 Rende, Cosenza, Italy; anita.giglio@unical.it (A.G.); brandmayr@unical.it (P.B.)

**Keywords:** Carabidae, sperm bundle protection, spermatostyle, cap, sperm ultrastructure, insect reproduction, ground beetle, male reproductive system

## Abstract

Ground beetle (Carabidae) spermatozoa are often aggregated in bundles of different sizes and are covered by a secretory material produced by the deferent duct epithelium. These secretions can have different shapes being either similar to a cap or to a rod (spermatostyle). Through a detailed study of the different regions of the male reproductive systems, we were able to describe the origin of this protective structure. We have obtained evidence that the secretions from the duct epithelium have initially an irregular shape, with the spermatozoa present in the duct lumen adhering to the structure with their heads on one side only. Later on, along their maturation, these structures bend, taking a cap shape. Hence, this process of secretion and successive evolution of the structures is very similar to that occurring in the spermatostyles. In conclusion, we obtained clear evidence that the two models of the sperm bundle protection share the same origin and thus they can be considered as homologous.

## 1. Introduction

Males of several insect orders have adopted efficient reproductive strategies to solve the problem of sperm competition at mating [1,2,3]. In many insects, females store in their spermatheca all sperm they receive from the partners with which they mate and sperm from a different male mix in the female storage organs, so that each contributing male may potentially fertilize part of the female’s clutch [4,5]. The male can bias paternity towards his own interests transferring at mating greater sperm quantities in an attempt to outnumber those stored from their rivals [3]. According to Parker [6,7], the production of a large number of tiny sperm is assumed to the best solution to the problem. The tendency to increase the sperm size, however, has an important adaptive significance because longer sperm have some advantages in the competition with shorter ones to fertilize eggs [7,8,9]. Moreover, in several insects, including ground beetles, males often protect sperm bundles with sperm heads embedded in a structure with a shape of a cap or a rod (spermatostyle) of variable size and length which is secreted by the epithelium of the deferent duct. During copulation, such sperm bundles are transferred to the bursa copulatrix of females, and from this district, they reach the spermatheca where they are stored to be used for fertilization [3,10,11,12,13]. It is suggested that sperm bundles facilitate the transfer of numerous sperm to females as they increase sperm mobility [14] and even give physical protection against spermicidal events [15]. Thus, the aggregation to form sperm bundles is a strategy largely adopted by numerous groups to increase the success in sperm competition and egg fertilization.

The type of sperm aggregations is quite variable, and according to Higginson and Pitnick [16], it can have a primary or secondary origin. In the first type, the sperm derive directly from the same spermatogonium present in the testicles while in the secondary type the sperm bundles are formed by units coming from different sperm cysts which join together during their transit along the deferent ducts. When such sperm aggregations are observed in the deferent duct, it is not simple to discriminate between the two types of sperm bundles.

The types of sperm aggregation described so far in different groups of Adephaga are as follows:(1)Individual sperm attached with their heads and the initial part of their flagella to a rod of extracellular material (spermatostyle). The sperm can be attached only along one side of the rod as it occurs in the Gyrinidae *Gyretes* sp. [17] or on whatever side as in *Harpalus* sp. [18] and other species of Carabids [12,19,20,21,22,23,24,25,26], in several species of *Dineutus* and in the whirligig *Orectochilus villosus* [27];(2)Sperm pairing when two sperm are associated by a fluffy secretion with their flat heads and the proximal tail region. This occurs in the water beetles *Dytiscus marginalis* and *Hydaticus seminiger* and in the diving beetle *Graphoderus* sp. [16,28];(3)Sperm associated in a small group by amorphous material as it occurs in the Colymbetinae (Dytiscidae) [29,30];(4)Sperm aggregates with heads and the anterior flagellar region embedded in an electron dense apical cap as it occurs in the *Carabus* species (Carabidae), in *Clinidium* and *Omoglymmius* (Rhysodinae), different in shape from the spermatostyle [23,24,26];(5)Sperm rouleaux when the tip of a sperm head slips into another sperm head to form a stack. This is made possible by apposition of one sperm cone-shaped side with the concave side of the successive sperm in the stack. This peculiar type of sperm aggregation has been described in some Dytiscidae Hydroporinae such as *Stictonectes optatus* [31] and *Hydroporus* sp. [16].

The different types of sperm aggregations can be adaptations due to the peculiar morphology of the sperm head, such as the above mentioned sperm rouleaux. However, in many other examples, it seems that they are due to the amount and elaboration of the secretion produced by deferent ducts epithelium along which the sperm run. This occurs in several ground beetles and also in the whirligig *Dineutus* ssp. and *Gyretes* sp. [17,18] where large secretions in the duct lumen take the shape of long rods on which the sperm are attached with their heads and a short anterior tract of the sperm flagellum [20,21,22,23,24,25]. In the Carabinae, however, the secretion forms a cap protecting the anterior sperm region.

Recently, we had the opportunity to study, at an ultrastructural level, the spermatogenesis and the secretion and successive differentiation of the apical cap protecting the sperm bundles in *Carabus lefebvrei*, an endemic Italian species. The results we obtained allowed confirming the presence of an apical cap protecting the sperm bundle, but also following the origin of such a cap. The collected evidence is indicative that the cap and the spermatostyle share a common origin even though they display different shapes at the end of their maturation.

## 2. Materials and Methods

Males of *C. lefebvrei* were hand-collected under rotten pine barks in the Sila National park (39°16′32.8″ N, 16°34′57.5″ E), San Giovanni in Fiore, Cosenza, Southern Italy in June 2023.

### 2.1. Fluorescence Microscopy

Bundles of sperm of *C. lefebvrei* were isolated from deferent ducts and seminal vesicles of adult males. The material was placed on a microscope slide in a small drop of 0.1 M phosphate buffer (pH 7.2) to which 3% of sucrose was previously added (PB). DNA was stained for fluorescence observations by incubating the sperm bundles for 3–4 min in Hoechst 33258 (1 μg/mL; Merck, Darmstadt, Germany). The sperm bundles were observed and photographed using interference contrast and epifluorescent microscopy with a Leica DMRB microscope equipped with AxioCam HR camera (Carl Zeiss, Oberkochen, Germany).

### 2.2. Scanning Electron Microscopy (SEM)

For the SEM study, *C. lefebvrei* sperm bundles from seminal vesicles were spread onto glass coverslips previously treated with 1% poly-L-lysine. The coverslips were placed in 2.5% glutaraldehyde in PB for 30 min at 4 °C and then rinsed several times in PB. Specimens on glass coverslips were dehydrated in a graded series of ethanol and then processed by the critical point drying method in a Balzer’s CDP 030 unit. The coverslips were sputtered with about 20 nm gold in a Balzer’s MED 010 sputtering device and finally observed in a Quanta 400 SEM (FEI Company, Hillsboro, OR, USA) operating at an electron accelerating voltage of 20 kV.

### 2.3. Transmission Electron Microscopy (TEM)

For TEM observations, adult males of *C. lefebvrei* were dissected in PB to isolate the testes and deferent ducts. The material was fixed overnight in 2.5% glutaraldehyde in PB. After careful rinsing, the material was post-fixed in 1% osmium tetroxide for 2 h. After rinsing, the material was dehydrated with ethanol series (50–100%), then transferred to propylene oxide and finally embedded in a mixture of Epon-Araldite epoxy resins. Semithin sections, obtained with a Reichert ultramicrotome, were stained with 0.5% toluidine blue and observed and photographed with a Leica DMRB light microscope equipped with an AxioCam Mrc5 HR camera (Carl Zeiss, Oberkochen, Germany). Ultrathin sections were routinely stained with uranyl acetate and lead citrate and observed in a TEM Philips CM10 operating at an electron accelerating voltage of 80 kV.

## 3. Results

### 3.1. Spermatogenesis and Spermiogenesis

As commonly observed in other insects and also in the species here examined, testes contain germ cysts in their apical region (Figure 1A). In each cyst (about 30 µm wide), undifferentiated spermatogonia (6–7.5 µm wide) are arranged in circles with their cytoplasm fused at the axial region and are surrounded by a thin cyst cell provided with an elliptical nucleus (9 µm × 3 µm) (Figure 1B). Spermatocyte cells (7.5–9.2 µm wide) are provided with randomly distributed mitochondria, a large nucleus and two orthogonally arranged centrioles (Figure 1C,D). The two centrioles are located close to the plasma membrane and often gave origin to two hemispheric plasma membrane protuberances (Figure 1C).

The spermatids, formed after the second meiotic division, are roundish (7.5–8.2 µm wide) (Figure 2A) and still maintain canal rings between them (Figure 1B,C). They have packed mitochondria (Figure 2B,C), forming a canonical “nebenkern” close to the elliptical nucleus. A short flagellum is visible emerging from the single centriole located close to the plasma membrane (Figure 2D).

The spermiogenesis proceeds with the formation of elongated flagellated cells (Figure 3B). They are provided with an apical bi-layered acrosome and an electron-dense nucleus (Figure 4B). At the end of spermiogenesis, the large sperm cyst have a huge lobate nucleus (9.0 µm wide and 3.0 µm thick) and a thin cytoplasm (only 1.2–2 µm thick) surrounding a sperm bundle (Figure 3A, Figure 4A–C and Figure 5).

In the inner region of the sperm cyst, depending on the sectioning level, numerous sperm heads and flagella are visible (more than 500 for each cyst, as a result of 2^9^ cell divisions). In other sections, the sperm show nuclei placed at the opposite sides of the cyst as a consequence of sperm cell looping (Figure 5). Longitudinal sections show the bi-layered acrosomes (only 0.6 µm long), the nuclei (about 10 µm long), and the long flagella (Figure 4B). Cross-sections through the sperm flagella show the common 9 + 9 + 2 microtubular axoneme, the two mitochondrial derivatives of the same size, and the two accessory bodies with that one on the right side elongated and greater than the opposite one (Figure 4D).

### 3.2. The Distal Deferent Duct (Vas Deferens I, Sensu [22])

Proceeding along the deferent duct, the cyst cells wall become fragmented, and sperm could mix with other ones from different bundles present along the duct lumen. It is not rare to find sperm bundles apparently with a different sperm number. Residual cytoplasm of the cyst wall with isolated nuclei and electron-dense bodies are visible in the periphery close to the epithelial cells of the deferent duct. These cells are 20 µm wide and have elliptical nuclei (8.2 µm × 4.1 µm) placed in their core region. The cytoplasm is very rich of spherical electron-dense inclusions. Apically, the cells are rich in mitochondria and small electron-dense vesicles (Figure 6A,B).

In the duct lumen, numerous sperm cells are visible together with elongated thick structures made of an electron-dense homogenous material. These structures are secreted by the epithelial cells. Cross-sections through the region just over the microvilli show electron-dense, elliptical, thin, short structures (some only 0.4–3.3 µm long)that adhered to others to give rise to long, fusiform formations (15.2–35.5 µm long and 1.1–2.2 µm thick) of electron-dense material (Figure 6C,D and Figure 7A–D). In a few sections, they look like large, flattened structures with a discoidal shape (Figure 7D). The observed variations in the lateral extension of the structures are presumably related to different sectioning levels of such discoidal masses of secretions. The secretion of these formations appears to follow a uniform sequence, since it is common to find several of them stratified over the epithelial cells (Figure 7A–D). In several points, it seems that a fusion between some of them occurred. As soon as the elliptical formations are secreted, the heads of the sperm present in the duct lumen come in contact with them and embed within these structures (Figure 6D and Figure 7A–D). When a series of these formations are in contact, their outer side, opposite to sperm heads, is marked by the presence of linearly parallel small electron transparent dots (Figure 7C). Once the longest formations detach from the apical microvillated epithelium, they descend along the duct lumen. It is clear that all the sperm are attached on one side only of the electron-dense formations. Some of these elongated structures take a curved appearance with the sperm protected underneath them (Figure 6D and Figure 7A). The dimensions of the curved elongated structures are quite variable; some of them measure with an about 13 µm length while others could reach up to a length of 18 µm and a thickness of 4 µm.

### 3.3. The Most Proximal Deferent Duct (Vas Deferens II, Sensu [22])

The sperm bundles, covered by the curved electron-dense formations, descend along the deferent duct. Some of the sperm bundles still maintain a similar appearance as they had in the previous level of the deferent duct (Figure 8A,C), but many others acquired a thicker and electron-dense apical protection. The appearance of these complexes is that of a giant cap, reaching up to 22.7 µm wide and 13.7 µm of thickness (Figure 8B,D). The appearance of the cap is, however, quite variable in shape and size, sometimes resulting in a long and thick rod (about 4.5 µm thick) to one side of which sperm are attached (Figure 8C). The increased dimension of the cap structure seems due to the apposition of a large amount of secretion by the epithelium surrounding the deferent lumen. Large masses of secretions are often visible, detaching from the apical epithelium region and stored beneath the cap of the sperm bundle, intermingled with the numerous sperm flagella (Figure 8D).

### 3.4. The Seminal Vesicles (Sensu [22])

Proceeding along the deferent duct, at the level of the seminal vesicles, the sperm cap of the sperm bundles may further increase in size reaching up to 42.2 µm of width, but the shape and the size of the numerous caps protecting the sperm bundles are quite variable and their thickness varies from only 3 µm to 12.2 µm (Figure 9A–E). This result seems to depend on the presence of a filamentous secretion in the duct lumen. These secretions originates from the numerous large secretory vesicles present in the apical epithelial region that further open into the lumen. The numerous sperm bundles and their caps present in the lumen result surrounded by a layer of these thin filaments (Figure 10A–C).

## 4. Discussion

In several ground beetles, the sperm in the male deferent duct can be free as it occurs for example in Cicindelinae, Scaritinae, Nebriinae, some Trechinae, Broscinae, Apotominae, and Paussinae [23,24,25,32]. However, the sperm of many other species of Carabidae are organized in bundles generated by a primary or secondary conjugation type mechanism (sensu [16]). Once arrived in the deferent ducts, the already assembled sperm bundles can host additional sperm cells from other bundles or lose some sperm. As a consequence, we could observe consistent size variations among the sperm bundles present in the deferent duct lumen. As described in [13], in several species of *Carabus*, an heteromorphism of sperm bundles is present. The presence of such a size variation in sperm aggregates was hypothesized to be an adaptation to minimize the risk of spermatophore displacement at mating by other competing males. It was also suggested that the observed size diversification of sperm bundles could be promoted by sexual selection via sperm competition with the larger sperm bundles being favored by their faster swimming performances [13]. Analogously, there is some evidence that sperm groups may be advantageous in post-copulatory sexual selection. In the fishfly *Parachauliodes japonicus* for example, it has been shown that large bundles containing more sperm reach the sperm storage organs more efficiently and faster than smaller sperm bundles [14].

Further on, the sperm bundles moving along the deferent duct receive a protective coating derived from secretions by the epithelial cells of deferent duct [12,19,21,22,23,24,25]. This protective secretion can be organized in different ways. It can be arranged in a cap structure as it occurs in some Carabinae and Rhysodinae [23,26] and possibly in other ground beetles [25], or in a rod structure, conventionally defined as spermatostyle as it occurs in several species belonging to different subfamilies of Carabidae [24,25] and also in whirligig beetles (Gyrinidae) [17,18].

The cup-like protection was initially considered as being generated by a mechanism different from that originating the rod-structure [24]. The present study, however, allowed for the clarification of how the bundle protection is originated and evolves in the examined ground beetle *C. lefebvrei*. A depth and accurate series of observations on the sequence of events occurring along male germ cells maturation from spermatids to sperm and then their association in bundles show that the secretions produced by the epithelial cells at the distal deferent duct give origin to elongated, end-pointed, thick electron-dense laminae. Our extended observations also show that sperm cells adhere to only one side of the laminae. Such completely unexpected evidence is confirmed by the identification of several sperm aggregations filling almost completely the deferent duct lumen.

The process of sperm adhesion to the secretion is very similar to what was previously observed in the whirligig beetle *Gyretes* [17]. In the case of *C. lefebvrei*, however, the process evolves further with the bending of the electron-dense laminae with adhering sperm. When such sperm bundles reach the successive proximal tract of the deferent duct, the region defined as vas deferens II [22], the cap structure protecting the sperm bundles become thicker by apposition of a further layer of secretions produced by the deferent duct epithelial cells. At this stage, the structure protecting the sperm bundle looks like a typical cap though with variations in size or in a few cases assuming bizarre shapes. As previously pointed out, a similar variation of the sperm protecting structures was described previously in several species of *Carabus* [13].

The evidence obtained in the present study are useful to clarify that the cap structure previously described as a different model of sperm aggregation from the spermatostyle [24] is actually produced in a quite similar way. In other words, although the final appearance of the two structures are apparently different, we now have evidence that the processes leading to the formation of both types of sperm bundles protection are similar, being characterized in both cases by the production by deferent duct epithelium of a secretions generating a structure to which the free sperm then adhere. The two organizations are generated with only one difference that we consider is not so relevant: in the spermatostyle rod, the sperm are able to adhere all around the structure, while they are seen adhering selectively to only one side of the laminae at the initial stage of adhesion to the structures which successively give origin to mature caps.

These results reconcile the ongoing discussions about the different types of sperm joining described in ground beetles [17,23,24,25]. The differences between the cap and the spermatostyle appear only at the final stages of their construction, but the two structures share the same origin and development and can therefore be considered homologous.

## Figures and Tables

**Figure 1 insects-15-00864-f001:**
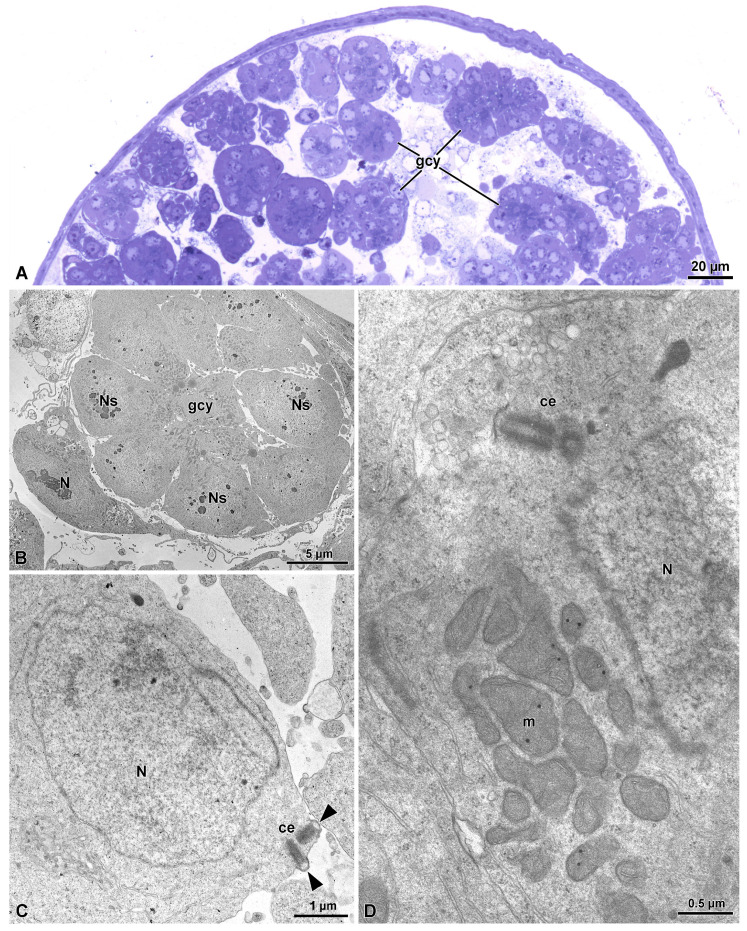
(**A**) Semithin section showing several clusters of germ cysts (gcy) from testis. (**B**) Cross-section of a germ cyst (gcy) with spermatogonial cells arranged in a circle. They are surrounded by a thin cytoplasm of the cell cyst. N, nucleus of the cell cyst; Ns, nuclei of the spermatogonia. (**C**,**D**) Spermatocyte with a large nucleus (N), the mitochondria (m), and the two orthogonally arranged centrioles (ce). Note in (**C**) arrowheads point to the small protuberances produced by the contact of centrioles with the plasma membrane.

**Figure 2 insects-15-00864-f002:**
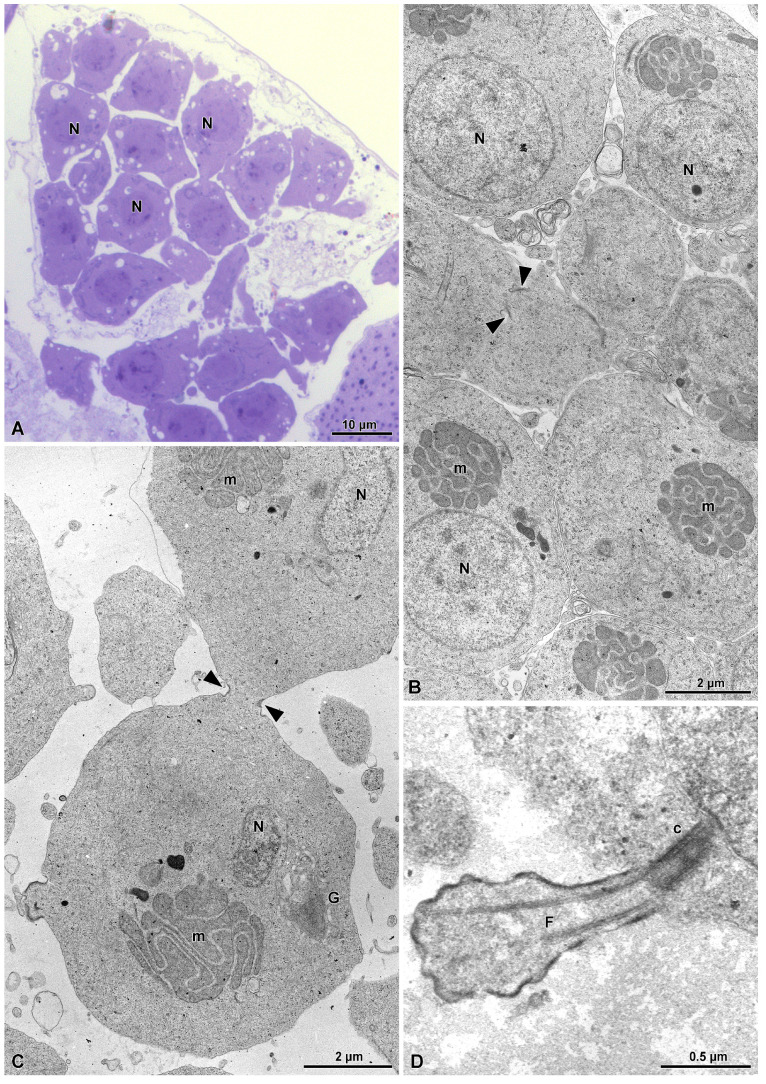
(**A**) Semithin section of a spermatid group. N, nuclei. (**B**) Cross-section through a spermatid group showing the nuclei (N) and the “nebenkern” of mitochondria (m). It should be noted that there is a ring canal (arrowheads) between two close spermatids. (**C**) Detail of a ring canal (arrowheads) between two spermatids. m, mitochondria forming a “nebenkern”; N, nuclei; G, Golgi complex. (**D**) Cross-section of the initial formation of the flagellum (F) from a single centriole (c) in an early spermatid.

**Figure 3 insects-15-00864-f003:**
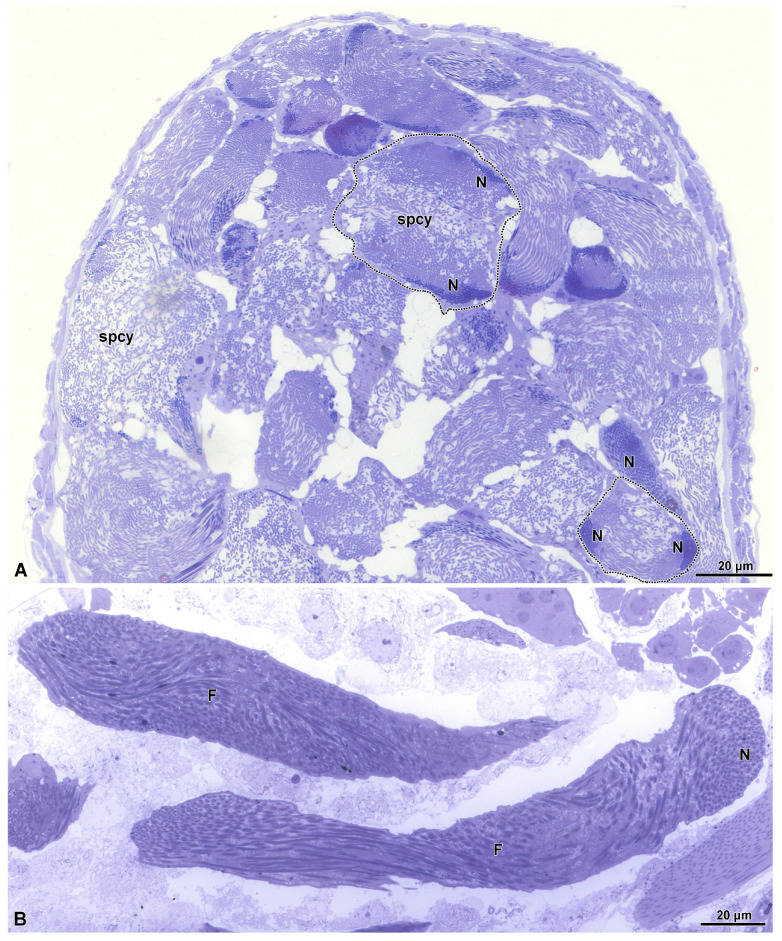
(**A**) Semithin section through a proximal testis region showing the numerous mature sperm cysts (spcy). The outlined cysts are illustrated in the micrograph of Figure 5. The dense regions of the section correspond to the nuclear position (N). (**B**) Longitudinal semithin section of a sperm cyst with the sperm nuclei (N) and their long flagella (F).

**Figure 4 insects-15-00864-f004:**
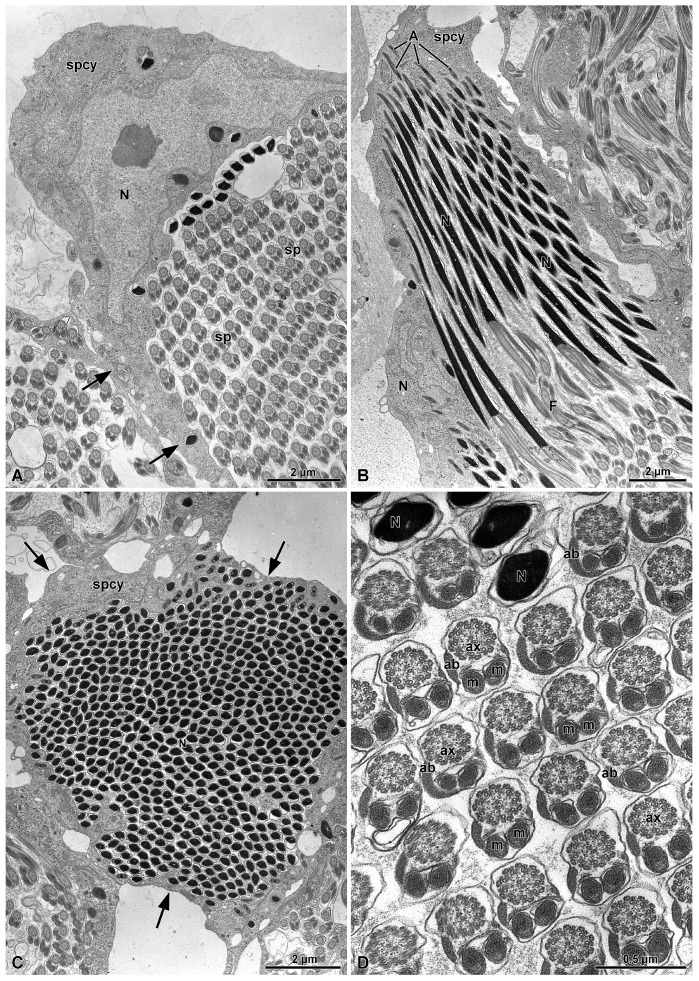
(**A**) Cross-section through a mature sperm cyst (spcy). The giant cyst cell have a huge nucleus (N) and a thin cytoplasm (arrows) surrounding the numerous cross-sectioned sperm (sp). (**B**) Longitudinal section of a mature sperm cyst (spcy) showing the acrosomes (A), the nuclei (N), and the flagella (F) of the numerous sperm. (**C**) Cross-section of a sperm cyst (spcy) at the level of the numerous nuclei (N), about 500 (indicated by the arrows). (**D**) Cross-section through the sperm of a sperm cyst. ax, axonemes; ab, accessory bodies; N, nuclei.

**Figure 5 insects-15-00864-f005:**
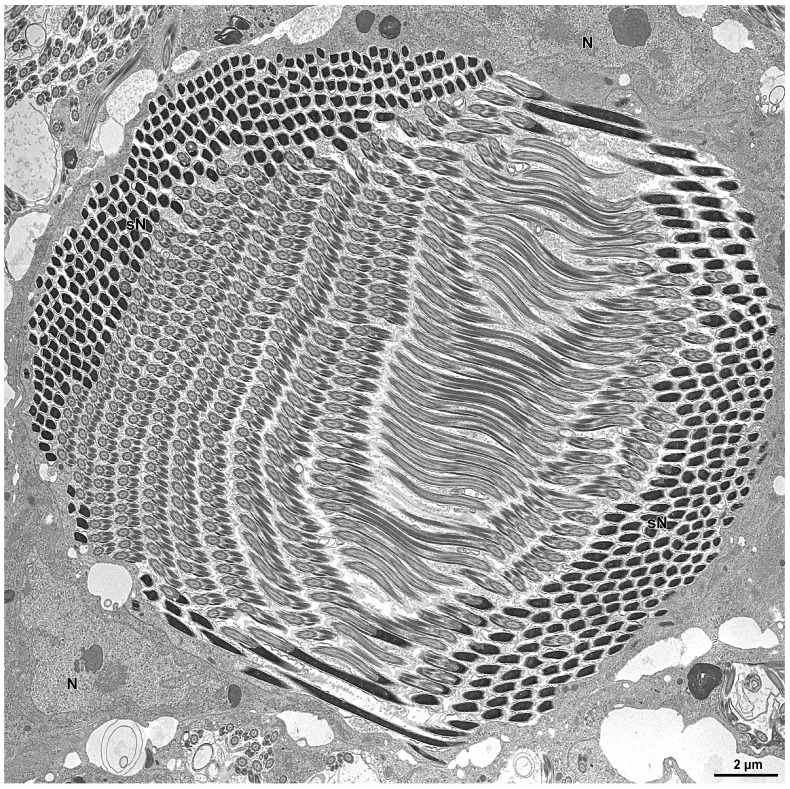
Compare the micrograph with the semithin section in Figure 3A. Cross-section through a mature sperm cyst surrounding the bundle of sperm. Due to the longer sperm flagella, a sperm looping occurs in the bundle, with the nuclei (sN) positioned at the opposite sides of the complex. Note the giant nucleus (N) of the sperm cyst.

**Figure 6 insects-15-00864-f006:**
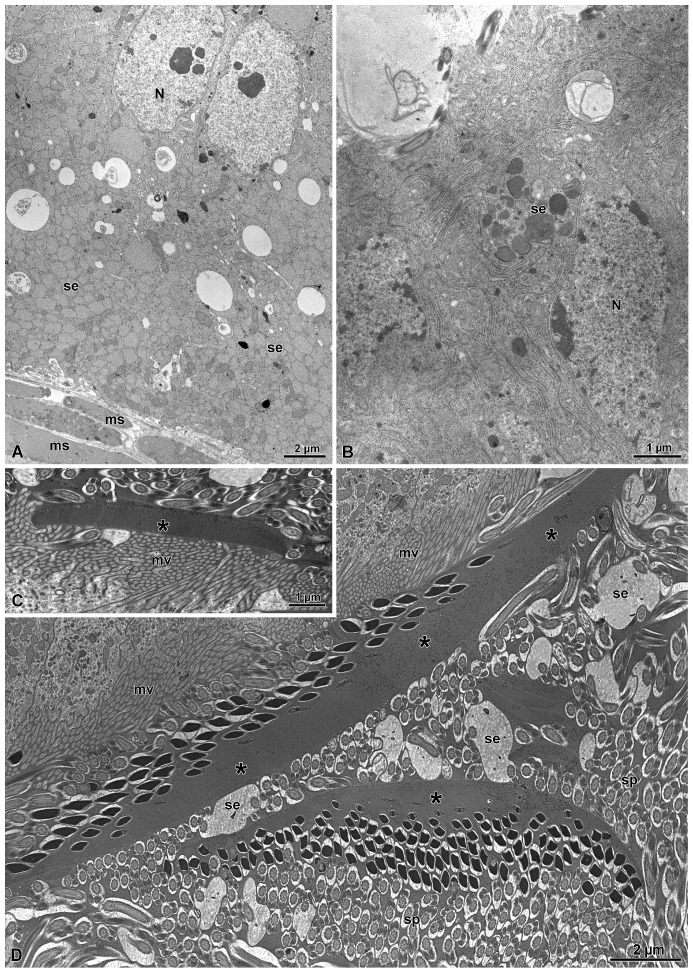
(**A**,**B**) Cross-section through the epithelial cells of the distal deferent duct. Note the numerous secretory vesicles (se) present in the cytoplasm. ms, basal muscle layer; N, nuclei. (**C**,**D**) Cross-sections of the apical epithelial region of the distal deferent duct showing the elongated electron-dense formation (asterisks) produced by the secretions (se) of the epithelial cells. Note that these structures are positioned just over the microvilli (mv), and as soon as they are detached from them, sperm (sp) present in the duct lumen adhered with their heads to them. The curved structure in the bottom of figure is also noted.

**Figure 7 insects-15-00864-f007:**
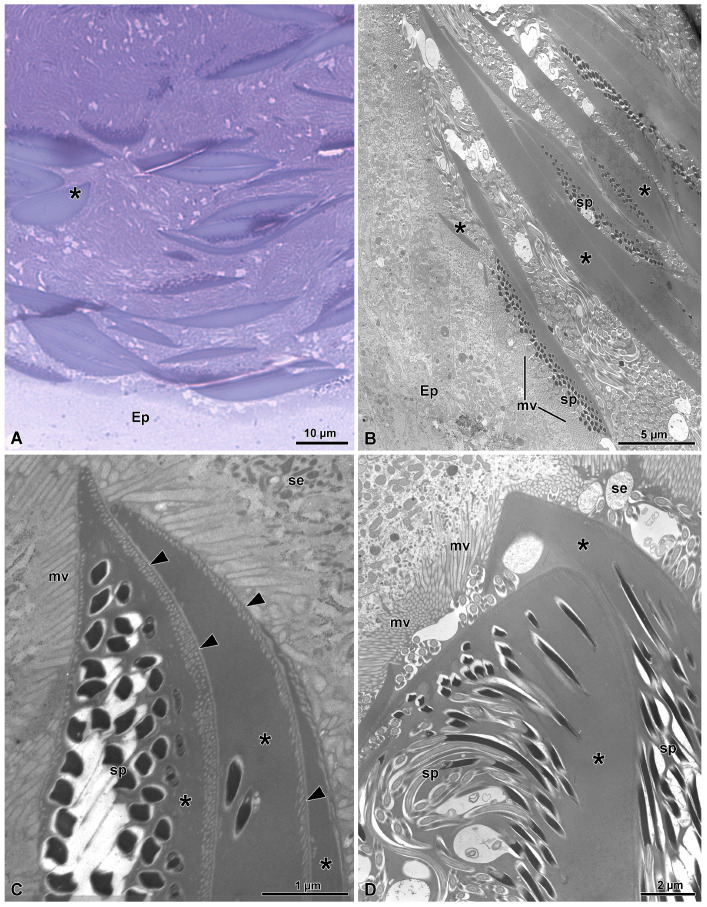
(**A**) Semithin section through the distal deferent duct showing several elongated structures secreted by the epithelial cells. These structures have a discoidal shape of variable thickness, depending on the section level. When a grazing section is observed (see on the left side of the figure), a flattened semicircular structure is visible (asterisk). Ep, epithelium. (**B**) Cross-section through the distal deferent duct showing the apical epithelial region (Ep) with microvilli (mv) over which several elongated structures of variable length are visible (asterisks). Along one side of them, sperm (sp) are attached. (**C**) Details of a few overlapped elongated structures (asterisks). Note the dot lines are along the periphery of the structures (arrowheads). mv, microvilli; se, secretions. (**D**) Detail of two overlapped elongated structures (asterisks). The one on the frontal position have a shape of a large flattened structure to which sperm (sp) are attached. mv, microvilli; se, secretion.

**Figure 8 insects-15-00864-f008:**
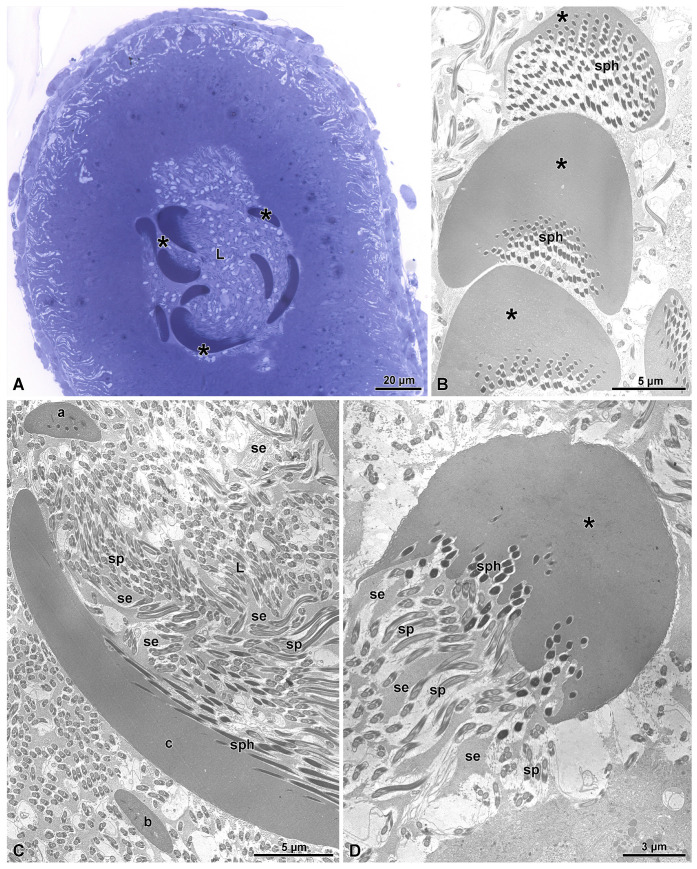
(**A**) Semithin section through the proximal deferent duct. In the lumen (L), several electron-dense structures cover the sperm bundles (asterisks). They have a variable size and shape. (**B**) Cross-sections of three structures covering the sperm bundles with a shape of large caps (asterisks). Beneath them, sperm heads (sph) are visible. (**C**) Cross-section of the duct lumen showing three structures (a, b, and c) of different sizes protecting the sperm bundles. The longest structure (c) shows the sperm heads (sph) adhering to only one side. In the lumen (L), many sperm flagella (sp) embedded in an electron-dense secretion (se) are visible. (**D**) Cross-section of a large cap structure (asterisk) showing the heads of the sperm bundle (sph) embedded in the structure. Note that sperm flagella (sp) are intermingled with an electron-dense secretion (se).

**Figure 9 insects-15-00864-f009:**
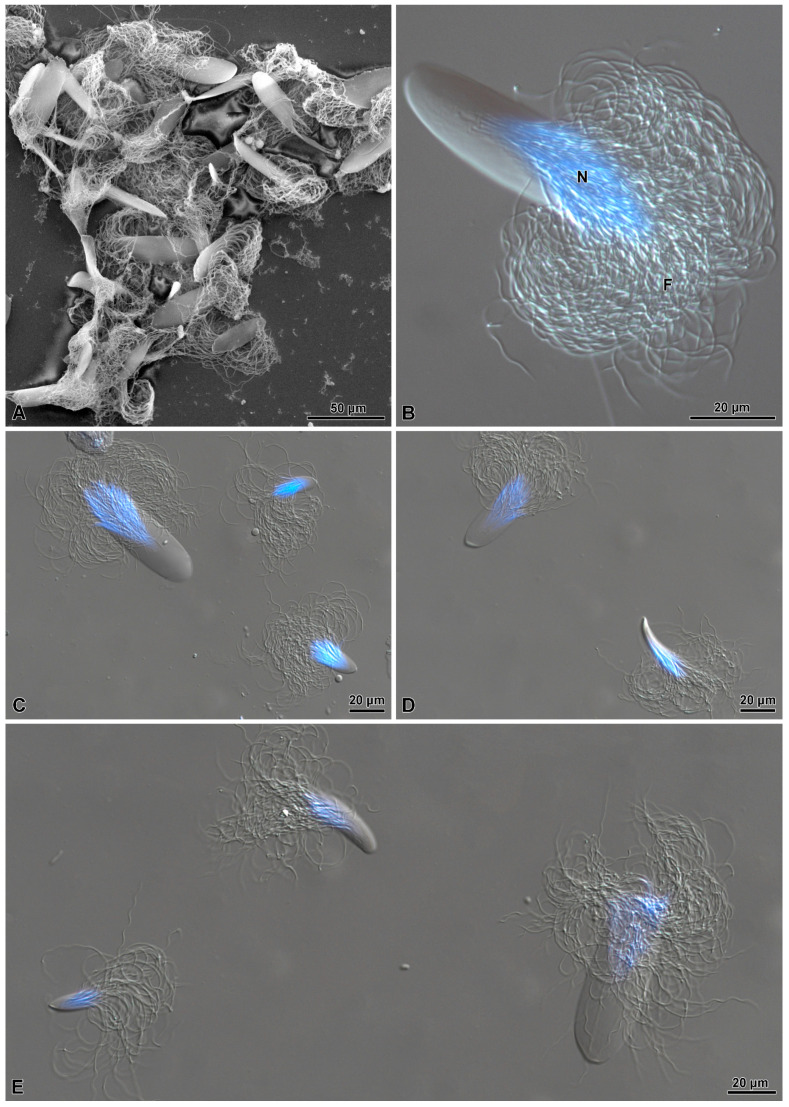
(**A**) SEM micrograph showing a compact group of sperm bundles with different shapes and sizes. (**B**) Hoechst staining of a giant sperm bundle with the apical conical protection from which part of the nuclei (N) (blue) and the flagella (F) emerge. (**C**–**E**) Hoechst staining of sperm bundles showing different shapes and sizes. Note, in (**C**,**E**), the smaller cup structures.

**Figure 10 insects-15-00864-f010:**
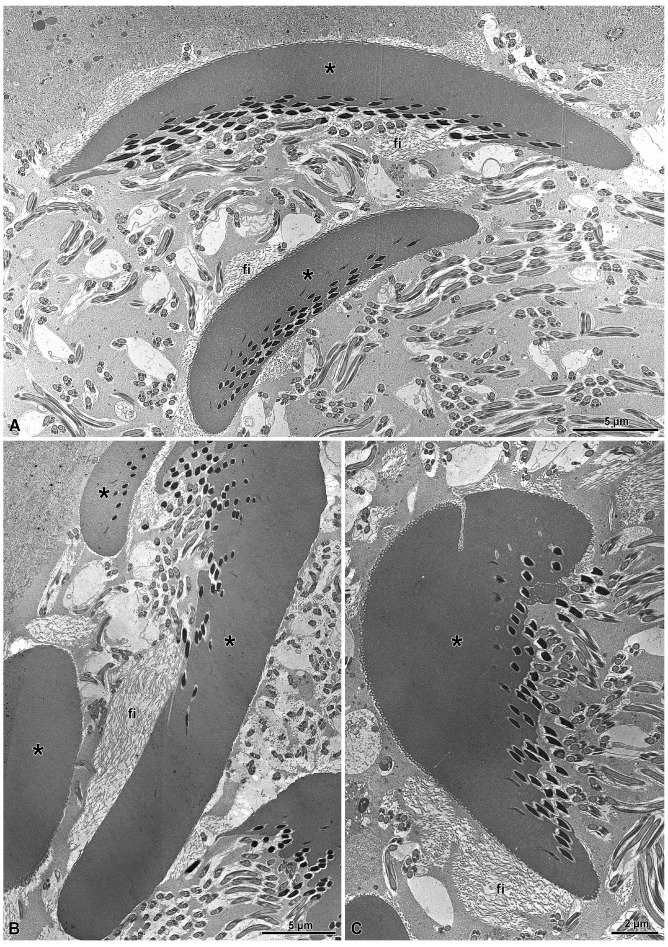
(**A**–**C**) Cross-sections through the lumen of the seminal vesicles showing differently shaped electron-dense structures (asterisks) protecting the sperm bundles. In (**A**,**B**), the structures are elongated and almost rod-shaped, while in (**C**) the shape is that of a giant cap. Large secretory vesicles containing filaments (fi) close to the structures are visible. Note that the sperm are adherent to the structures only along one side.

## Data Availability

The original contributions presented in this study are included in the article. Further inquiries can be directed to the corresponding author.

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
