# Peer review of "The Cap and the Spermatostyle Protecting the Sperm Bundle Have a Similar Origin—Ultrastructural Study of the Spermatogenesis from the Ground Beetle Carabus (Chaetocarabus) lefebvrei Dejean, 1826 (Adephaga Carabidae)"

_insects, 2024, doi:10.3390/insects15110864_

Round 1

Reviewer 1 Report

Comments and Suggestions for Authors

This paper is interesting, cogent, well-illustrated, well-written, and contains novel material. The organization is logical, conclusions are justified, all the illustrations provide important information, and the references are appropriate. I recommend publication with just the few minor changes that I suggest below, and editing for a few minor English grammatical errors that do not affect comprehension.

l. 166 See comment about line 190

l. 190 Fig. 4A is an electron micrograph. Do you mean Fig. 3A? In any case, it is hard to delineate the boundaries of the cysts in Fig. 3A, so I suggest drawing outlines around the two labeled cysts.

l. 193 I find it hard to visualize the “looping”; a simple diagram of the looped cyst would help.

l. 204 Distal to what? I think you mean relative to the gonopore (aedeagus), so say that.

l. 236 Fig. 7B needs a micrometer scale

l. 243 I recommend using the term ”covered” instead of “protected”, as the latter implies a function, which is unknown. Apply in other parts of the paper, e.g. in the legend of Fig. 8.

l. 265 For clarity, after “…deferent duct,” insert “termed by Schubert et al (2017) as the seminal vesicles,”

Author Response

Comment 1: l.166 See comment about line 190

Comment 2: l.190 Fig. 4A is an electron micrograph. Do you mean Fig. 3A? In any case, it is hard to delineate the boundaries of the cysts in Fig. 3A, so I suggest drawing outlines around the two labelled cysts.

Response 1 and 2: We have corrected Fig. 4A in Fig. 3A as indicated. In the semithin section of Fig. 3A we have outlined the cysts similar to those in Fig 5.

Comment 3: l.193 I find it hard to visualize the “looping”; a simple diagram of the looped cyst would help.

Response 3: The “looping” is the result of bending of very long sperm when they cannot be contained within a cyst cell with a limited volume (Syed et al., Cells 2021, 10, 2762). In the micrograph, the nuclei are in opposite sides and the flagella in the middle region. This means that the sperm are disposed in two opposite regions of the cyst due to their high number and length.

Comment 4: l.204 Distal to what? I think you mean relative to the gonopore (aedeagus), so say that.

Response 4: The term distal and proximal deferent duct means that the deferent region corresponds to the vas deferens I and vas deferens II as described by Schubert et al. (PloS one 2017, 12(7), They have the same meaning of close to the testes (distal) or to the seminal vesicles and ejaculatory duct (proximal), respectively.

Comment 5: l.236 Fig. 7B needs a micrometer scale

Response 5: We have added the micrometer scale in Fig. 7B.

Comment 6: l.243 I recommend using the term “covered” instead of “protected”, as the latter implies a function, which is unknown. Apply in other parts of the paper, e.g. in the legend of Fig. 8.

Response 6: We have corrected the term “protected” in “covered” as requested

Comment 7: l.265 For clarity, after “…deferent duct,” insert “termed by Schubert et al (2017) as the seminal

Response 8: We have added at line 265 the citation as request. We have added a sentence at line 266 to indicate that the last increased structure of the sperm cap of the sperm bundles occurs in the seminal vesicle.

Reviewer 2 Report

Comments and Suggestions for Authors

This paper uses electron microscopy and other techniques to observe the state of sperm ultrastructurally in various parts of the male reproductive organs of Carabus.
The results show that in this insect, the structures defined as the cap and rod (spermatostyle), which form sperm aggregates as the sperm mature, are different objects, but their origin is the same.
The experimental method and presentation of the results are clear and of sufficient quality to be published in this journal.

However, the following points should be addressed.The English is not well written, so it is difficult to understand.
1. The English is not well written in general, so it is difficult to understand.
2. it would be better to include pictures so that the reader can understand what kind of insects they are.
3. I would like to see a picture or a schematic diagram of the male reproductive organs. When reading the text, it is extremely difficult to understand which part of the sperm was taken out and from which part of the sperm.
This paper uses electron microscopy and other techniques to observe the state of sperm ultrastructurally in various parts of the male reproductive organs of Carabus.
The results show that in this insect, the structures defined as the cap and rod (spermatostyle), which form sperm aggregates as the sperm mature, are different objects, but their origin is the same.
The experimental method and presentation of the results are clear and of sufficient quality to be published in this journal.

However, the following points should be addressed.The English is not well written, so it is difficult to understand.
1. The English is not well written in general, so it is difficult to understand.
2. it would be better to include pictures so that the reader can understand what kind of insects they are.
3. I would like to see a picture or a schematic diagram of the male reproductive organs. When reading the text, it is extremely difficult to understand which part of the sperm was taken out and from which part of the sperm.
4. A graphical schematic of the changes in sperm status during sperm maturation would clarify what is novel in this paper.

Comments on the Quality of English Language

It was difficult to understand the whole text, probably due to the quality of the English.

Author Response

Comment 1: The experimental method and presentation of the results are clear and of sufficient quality to be published in this journal.

Response 1: It is pity the Reviewer has weakly appreciated the quality of the numerous figures accompanying the text. We have, however, a different opinion!

Comment 2: The English is not well written in general, so it is difficult to understand.

Response 2:  The English was reviewed by a colleague who knows well the language.

Comment 3: It would be better to include pictures so that the reader can understand what kind of insects they are.

Response 3: The carabid insects are quite common species and it is not necessary to illustrate the habitus of the species, as it is well known by specialists of the group.

Comment 4: I would like to see a picture or a schematic diagram of the male reproductive organs. When reading the text, it is extremely difficult to understand which part of the sperm was taken out and from which part of the sperm.

Response 4: The schematic organization of the male reproductive system is similar to that described by Schubert et al. (PloS one 2017, 12(7), as indicated in text paragraphs.

Comment 5: A graphical schematic of the changes in sperm status during sperm maturation would clarify what is novel in this paper.

Response 5: The figures of semithin sections and of micrographs at low magnification, clearly show the different stages of spermatogenesis occurring in the testes and spermiogenesis. We therefore think it is not necessary to add new figures to the work, also considering the numerous plates already accompanying the text.

Round 2

Reviewer 2 Report

Comments and Suggestions for Authors

This paper uses electron microscopy and other techniques to observe the state of sperm ultrastructurally in various parts of the male reproductive organs of Carabus (Chaetocarabus) lefebvrei.

The authors found that during the early stages of cap formation, spermatozoa attach to only one side of the spermatocyst, which is different from their attachment along the bars of the spermatostyle. In this respect, they find that the cap structure is similar to the protective structure found in the spermatozoa of several species of whirligig beetle.

The results show that in this insect, the structures defined as the cap and rod (spermatostyle), which form sperm aggregates as the sperm mature, are different objects. Still, their process of secretion and successive evolution of the structures is the same.  The experimental methods and presentation of results supporting the above scientific presence are clear and of sufficient quality to be published in this journal.